# Bell’s Palsy and COVID-19 Vaccines: A Systematic Review and Meta-Analysis

**DOI:** 10.3390/vaccines11020236

**Published:** 2023-01-20

**Authors:** Khaled Albakri, Abdulrhman Khaity, Hany Atwan, Othman Saleh, Momen Al-Hajali, Shirin Cadri, Rehab Adel Diab, Ebraheem Albazee, Ahmed Negida

**Affiliations:** 1Department of Surgery, Faculty of Medicine, The Hashemite University, Zarqa 13133, Jordan; 2Medical Research Group of Egypt (MRGE), Cairo 11511, Egypt; 3Department of Surgery, Faculty of Medicine, Elrazi University, Khartoum 1115, Sudan; 4Department of Surgery, Faculty of Medicine, Assiut University, Assiut 71515, Egypt; 5Department of Dermatology, Faculty of Medicine, Grigore T. Popa University of Medicine and Pharmacy, 700115 Lasi, Romania; 6Department of Surgery, Faculty of Medicine, Al-Azhar University, Cairo 11651, Egypt; 7Department of Internship, Kuwait Institute for Medical Specializations (KIMS), Kuwait City 13109, Kuwait; 8Department of Surgery, Faculty of Medicine, Zagazig University, Zagazig 44519, Egypt; 9Department of Global Health, Harvard Medical School, Boston, MA 02115, USA

**Keywords:** COVID-19, vaccine, Bell’s palsy, side effect, systematic review

## Abstract

Introduction: Once the COVID-19 pandemic was declared, the world was waiting for the clue that would be able to cut down the progression of the disease. Vaccines play a crucial role in reducing the disease and saving many people worldwide. However, there are several side effects of these vaccines, like pain, fatigue, fever, and neurological defects like Bell’s palsy. In this systematic review, we presented evidence about the occurrence of Bell’s palsy followed by COVID-19 vaccination. Methods: We searched PubMed, SCOPUS, EBSCO, and Web of Science (WOS) from inception till October 2022. The quality assessment was conducted using the Joanna Briggs Institute, National Institute of Health, and Newcastle-Ottawa. The analysis was conducted on SPSS. Results: Thirty-five records were involved in our study. The results of our cases revealed that most of the patients (62.8%) experienced unilateral facial paralysis. Also, the majority of the cases were reported after the first dose, and most cases were after Pfizer, AstraZeneca, and Sputnik V vaccines, respectively. The patients who were treated with corticosteroids, IVIG, and anti-viral drugs, showed marked recovery afterward. Conclusion: The rate ratio of Bell’s palsy after COVID-19 vaccination was 25.3 per 1,000,000. The ratio was higher after the first dose compared to the second dose and was higher among those who took Oxford/AstraZeneca vaccine compared to other vaccines. However, this condition was reported in a small number of cases among a large number of vaccinated people worldwide. It is important to note that the benefits of getting vaccinated far outweigh any potential risks.

## 1. Introduction

Once the COVID-19 pandemic was declared, the whole world was waiting for the clue that would be able to cut down the progression of the disease. Vaccines play a crucial role in reducing the disease and saving many people worldwide. As anything could have a beneficial and harmful effect and still be used as the benefits outweigh the risks, Vaccines used but underwent multiple trials to detect safety and any adverse events trying to help people in the decision regarding using the vaccine [1]. Literature showed a wide spectrum of reported complications associated with the vaccines, either neurological, cardiovascular, cerebrovascular, etc. [2,3].

This study will deal with one of the rare neurological complications associated with different types of COVID-19 vaccination, Bell’s palsy and other disorders that mimic facial palsy. Facial nerve paralysis can be reported as isolated or a part of the autoimmune disease developed with the vaccine as Guillain-Barre syndrome (GBS), Polyneuritis or other forms of autoimmune neuropathy [4]. Bell’s palsy is an idiopathic paralysis of the facial nerve with rapid onset and underdetermined etiology diagnosed by exclusion of other causes of facial nerve paralysis. Affect population with age ranging from 15 to 50 years old and able to affect males and females equally [5].

Facial paralysis associated with Bell’s palsy, if untreated, can be spontaneously resolved within 2 months by 70%, but if managed with corticosteroids, the percentage of resolution is elevated to 90% [5]. Facial palsy shows reported to be 25 cases per 100,000 per year in the general population, and Yellow card reporting systems in the United Kingdom (UK) have recorded 291 cases of facial paralysis with the Pfizer vaccine from 9 December 2020 to 5 April 2021. Although it is shown to be high, the MHRA says it is not higher than that of the general population [6]; therefore, continuous monitoring of the side effects is important. The three major vaccines in the UK, Moderna, AstraZeneca and Pfizer/BioTech, reported facial nerve palsy in their initial trials [7,8]. In the phase 3 trial of the Pfizer/BioTech vaccine, over 38,000 patients showed four cases of Bell’s palsy compared to zero reported cases in the placebo group [8]. The UK government’s website considered facial paralysis as a potential side effect of the Pfizer/BioTech vaccine; therefore, reporting cases of facial paralysis associated with the Pfizer/BioTech vaccine is important to support or exclude this theory [8].

In the phase 3 trial, safety analysis of the Moderna vaccine conducted over 30,420 patients also reported four cases of Bell’s palsy, three from the vaccine group and only one from the placebo group, while the Oxford/Astrazeneca vaccine trial conducted in over 23,745 participants reported six cases, three from vaccine group and three from the placebo group [9]. Therefore, monitoring cases of facial paralysis associated with different types of vaccines is important to detect the possible relationship between the vaccine and facial palsy. All reported cases showed the association without any causal relationship, and the U.S. Food and Drug Administration (FDA) recommends continuous monitoring to detect any possible side effects associated with the vaccines. Our study aimed to conclude and analyze the available evidence in the literature to help the researches to formulate the next research questions and continue monitor of the associated side effects with different types of the vaccine and even monitor unilateral facial nerve palsy and also help the clinicians to take the decision regarding the vaccine administration to different population taking in consideration the risks and benefits.

Recent studies suggested further evaluation of bell’s palsy and COVID-19 vaccination, and till now, there has been no strong evidence that can confidently conclude whether there is an association between bell’s palsy and COVID-19 vaccinations or not [10,11]. Furthermore, a systematic review of 17 studies was published to investigate the existence and development of COVID-19 symptoms after vaccinations [12]. They concluded that COVID-19 vaccination before SARS-COV-2 infection could reduce the risk of long-term symptoms of COVID-19.

Therefore, our aim is to specifically and comprehensively pool all records that reported the occurrence and characteristics of Bell’s Palsy after COVID-19 vaccinations.

## 2. Methods

We followed the guidelines of the Preferred Reporting Items for Systematic Review and Meta-analysis (PRISMA) when conducting this systematic review and meta-analysis. Also, this investigation was registered in the International Prospective Registration of Systematic Reviews (PROSPERO) database; [registration ID: CRD42023389393].

### 2.1. Literature Search Strategy

We searched PubMed, SCOPUS, EBSCO, and Web of Science from inception till October 2022. Our search strategy comprised: [ (COVID-19 OR “SARS-COV-2” OR “corona virus-19” OR “COVID-19 vaccines” OR “SARS-COV-2 vaccines” OR “ChAdOx1 nCoV-19 vaccine” OR “Astra- 89 Zeneca Vaccine” OR “Ad26.COV2. S” OR “Johnson & Johnson Vaccine”) AND (“bell palsy” OR “facial paralysis” OR “bell’s pals*” OR “acute inflammatory facial neuropathy)]. To broaden the literature search, we scanned the reference lists for eligible studies and contemporary reviews for potentially missed records. The study selection process comprised omitting duplicate records, followed by TITLE/ABSTRACT screening, then concluded by FULL-TEXT screening. Two independent co-authors completed this task, and any disagreements were solved by the primary investigators.

### 2.2. Eligibility Criteria and Study Selection

Studies achieving our inclusion and exclusion criteria were included: we included studies that reported (i) patients underwent any kind of COVID-19 vaccination, (ii) both genders and all age groups, (iii) the incidence and characteristics of bell’s palsy after COVID-19 vaccination, (iv) with the following study designs: case reports and case series, observational studies, and clinical trials. We excluded (i) animal, laboratory, in vitro studies, conferences, reviews, and book chapters and (ii) studies that did not mention bell’s palsy as an event.

### 2.3. Data Extraction

We extracted the data from each included study using a pre-specified uniform data extraction sheet. The extracted data included the following domains: study design, the number of included patients, age, sex, comorbidities, main signs and symptoms, the type of vaccine received and the number of doses, onset of fascial palsy, diagnostic tools, treatments, follow up duration and prognosis.

### 2.4. Quality Assessment

We have implanted the Joanna Briggs Institute (JBI) [13] and National Institute of Health (NIH) [14] quality assessment tools for our included Case reports and case series, respectively, based on the clinical characteristics, history, diagnostics, interventions and management plans with evidence score ranging from Poor, Fair or Good. Newcastle-Ottawa (NOS) [15] tool for included observational studies Cohort, Case-control and Cross-sectional studies based on selection, comparability and Outcome domains. Quality assessment is done aiming to estimate the overall evidence that our study deals with. Two Authors independently assessed the evidence and to remove the conflicts.

### 2.5. Statistical Analysis

The statistical analysis was conducted on SPSS. Number and percentages were used to describe the categorical variables, as well as mean and standard deviation (SD) was used to describe the continuous variables.

## 3. Results

After searching different databases, we obtained 430 records. After removing the duplication, 305 records underwent the first screening. Of them, 52 reports were included. (Figure 1.)

This systematic review included cohort (*n* = 13), case-control (*n* = 3), cross-sectional studies (*n* = 3) and self-controlled case series (*n* = 2) that described the rate ratio of Bell’s palsy after administration of COVID-19 vaccines. Further, this review included case reports (*n* = 23) and case series (*n* = 8), which described patients developing events of Bell’s palsy after COVID-19 vaccination (*n* = 105). The majority of cases were men (57.1%). The age of the patients varied from approximately 30–60 years; the average age of patients was 49.7 ± 23.2 years. The most common comorbidity among the included patients was hypertension (24.8%), followed by diabetes (16.2%) and heart failure (4.8%). According to the administrated vaccines, 49 patients took Pfizer (46.7%), while 22 (21%) and 16 (15.3%) of the cases took Sinovac Biotech and AstraZeneca, respectively. Most cases developed the symptoms after the first dose of vaccines (35.2%). The average time to symptom onset after the vaccination was 11.6 days. The majority of cases took prednisolone (38.5%). Regarding the prognosis, 36 cases (69.2%) have recovered (26 completely and 13 partially). More details can be seen in Table 1. The main clinical and pathological characteristics of the patients included in our study were summarized in Appendix A. The characteristics of the cohort, case-control, cross-sectional studies, and self-controlled case series included in this review were presented in Appendix A.

According to Table 2, the percentage of recovered females was slightly higher than recovered males (92.9% and 88.5%, respectively). All the cases who developed Bell’s palsy after the second dose recovered, while only 90% recovered among those who developed this condition after the first dose. Non-hypertensive individuals recovered at a higher rate (100%) than hypertensive ones (50%).

According to Figure 2, the rate ratio of Bell’s palsy among vaccinated groups was 25.3 per 1,000,000. The ratio was higher after the first dose compared to the second dose (Rate Ratio = 171.2 and 42.1, respectively). Patients who took the Oxford/AstraZeneca vaccine revealed higher ratios (Rate Ratio = 178.6) compared to other identified vaccines.

### Quality Assessment

The overall quality assessment of the 24 included case reports based on the JBI checklist ranged from moderate to high evidence, with 17 case reports showing a low risk of bias and seven studies with a moderate risk of bias supporting their inclusion based on JBI recommendations, as shown in Table 3. The risk of bias assessment of the seven included case series based on NIH scores revealed six studies with fair-quality evidence and one with good evidence, as shown in Appendix A. The assessment of the included 15 cohort studies showed eight studies with good evidence, one with fair evidence and seven with poor evidence based on the NOS scale modified for cohort studies, as shown in Appendix A. The included seven case controls show one with fair evidence and six with high-quality evidence based on the NOS scale, as shown in Appendix A. All the included cross-sectional studies show good-quality evidence, as shown in Appendix A.

## 4. Discussion

### 4.1. Summary of the Findings

An increasing incidence of neurological disorders has escalated following COVID-19 vaccine administrations among the general population [38]. Bell’s palsy is one of these neuropathies that may have a close relationship with COVID-19 vaccines [39]. The results of our cases revealed that most of the patients (62.8%) experienced unilateral facial paralysis. The majority of the cases were reported after the first dose, and most cases were after Pfizer, AstraZeneca and Sputnik V vaccines, respectively. According to the diagnostic measures, the physical examination, MRI, and lumbar puncture were the most used in the included studies. The patients who were treated with corticosteroids, IVIG, and antiviral drugs, showed marked recovery afterward. Hypertension was the most common comorbidity among the included patients. In addition, there was a statistically significant difference between prognosis and blood pressure status.

The exact mechanism of developing facial nerve palsy following COVID-19 vaccination is yet to be unclear [40]. Multiple theories suggest the possible pathophysiology behind this phenomenon. One theory suggests the involvement of autoimmune processes either via mimicry of host molecules by the vaccine antigens or by bystander activation of autoreactive dormant T-cells [17,41]. Moreover, a study regarding the immunological activities after COVID-19 vaccine administration showed that a significant increase of antibody titers occurs following the two doses of the BNT162b2 vaccine [42]. This huge stimulation of the immune system might as well contribute to the development of Bell’s palsy [43]. Another theory states that the reactivation of different dormant viruses in the CNS might cause palsy. These viruses include the varicella-zoster virus, the Usutu virus, human herpesvirus 6 and herpes simplex virus type 1 (HSV-1). It is understood that perturbation of type I IFN signaling could lead to reactivation of the latent herpes simplex infection in neurons near the geniculate ganglion, which may prompt facial palsy (3;4). When type I IFN signaling is flawed, as it is done after vaccination with SARS-CoV-2, the capability of CD8+ T cells to retain herpes under control would also be damaged. The reactivation of HSV-1 is probably the most accepted theory [33]. A study involving 14 patients with Bell’s palsy, nine with Ramsy–Hunt syndrome, and 14 control patients, managed to show a link between the presence of the HSV-1 genome in patients and the development of Bell’s palsy [17]. In addition, it is worth mentioning that the influenza vaccine has been previously linked to an increased incidence of Bell’s palsy. The mechanism of this was also thought to be the reactivation of latent HSV-1 in the geniculate ganglion of the facial nerve, which in turn caused inflammation of the nerve [16,44]. Based on several hypotheses, the interaction between elements of the vaccine and specific human proteins’ immune cross-reactivity consequences in autoimmune disease, which may cause Bell’s palsy [45]. Innate immunity could be activated by the vaccine’s mRNA and lipids, which induce the synthesis of interferons and possibly damage the myelin sheath [41,46]. This theory was further supported by the fact that interferon therapy in hepatitis C infection patients indeed showed a rare but evident occurrence of Bell’s palsy [46,47,48]. Nevertheless, it is important to mention that Bell’s palsy can also happen secondary to GBS or other demyelinating disorders since these disorders are rare but possible adverse effects following COVID-19 vaccination [40,43]. After all, we can say that the pathophysiology of such a phenomenon could be caused by multiple factors at once and may differ from person to person, given the fact that most reported cases of Bell’s palsy after COVID-19 vaccination either show differences in the time of onset or the severity of symptoms presented [40,43].

In previous systematic reviews conducted by Shahsavarinia et al. [10] and Khurshid et al. [49], the most notable comorbidities identified among these studies were hypertension followed by diabetes and dyslipidemia, which is consistent with our study. At the same time, in the current study, we conducted the analytical test to determine the discrepancy between the variables and prognosis. Although there was no statistically significant difference concerning the prognosis and most of the comorbidities, patients without hyperuricemia or hypertension had a high rate of full recuperation in comparison with patients with these diseases. Our findings are in the same direction as the former reviews performed by Sriwastava et al. [50] and Shahsavarinia et al. [10] in terms of the association between the type of vaccines and the incidence of facial palsy. Unlikely, Khurshid et al. [49] reported that most of the cases had bell’s palsy after taking the AstraZeneca vaccine. This could be elucidated by the variation in sample size and baseline characteristics. However, our results are in agreement with Khurshid et al. [49] in the case that the majority of symptoms have appeared following the first dose of the vaccines.

The average time from vaccination to onset of symptoms amongst the cases in our review was 11.49 (11.064) days. This is consistent with previous studies by Shahsavarinia et al. [10] that mentioned the onset of symptoms of bell’s palsy at around 10 days post-vaccination. On the other hand, our study represented that (62.8%) of the patients had unilateral-side complaints. As evidenced by Khurshid et al. [49], most of the patients (53.45%) presented with bilateral facial weakness.

Diagnosis of any facial palsy cases is often based on physical examination and is confirmed through electromyography, lumbar puncture, CT, and MRI. In our systematic review, (95.35%) of the cases were diagnosed through physical examination, (67.44%) using MRI, and (48.84%) by lumbar puncture. In contrast, Khurshid et al. [49] study documented the majority of the included patients were diagnosed by CSF analysis and then by CT and MRI. This can be attributed to the fact that most participants in this study may have different characteristics from Khurshid et al. study [49]. Regarding the management, (46.51%) and (32.5%) of the patients in our study who received corticosteroids and IVIG therapy, respectively, have more quick mitigation of the symptoms. This result was also identified in another systematic review by Khurshid et al. [49].

Generally, the incidence of neurological diseases after vaccination varies depending on demographic features, residual confounding, type of vaccines, and reporting bias that may have a potential impact on the consequences [51]. It is critical to highlight that the causality relationship between Bell’s palsy and COVID-19 vaccines is not confirmed yet. Therefore, we recommend conducting further studies in an attempt to conclude the possibility of synergistic influence between facial palsy and COVID-19 vaccines.

### 4.2. Limitations and Strengths of the Study

The major limitations in this study have been related to a limited number of included studies that assessed the association between facial palsy and COVID-19 vaccines. Hence, the small sample size, which is reflected in the strength of the evidence. Nevertheless, the strengths of our study are as follows: (1) our systematic review provided the analytical investigation between the variables and prognosis of patients with bell’s palsy after commencing the vaccination, (2) we provided a more comprehensive analysis in an attempt to summarize the probability association between COVID-19 vaccines and facial palsy.

## 5. Conclusions

The rate ratio of Bell’s palsy after COVID-19 vaccination was 25.3 per 1,000,000. The ratio was higher after the first dose compared to the second dose and was higher among those who took the Oxford/AstraZeneca vaccine compared to other vaccines. The average time to symptom onset after the vaccination was 11.6 days. Hypertension was the most common comorbidity among the included patients. Moreover, prednisone was the most frequent drug used to manage this condition. Overall, the majority of the patients recovered. However, this condition was reported in a small number of cases among a large number of vaccinated people worldwide. Additionally, it is important to note that the COVID-19 vaccine has been shown to be safe and effective in preventing illness as well as the benefits of getting vaccinated far outweigh any potential risks.

## Figures and Tables

**Figure 1 vaccines-11-00236-f001:**
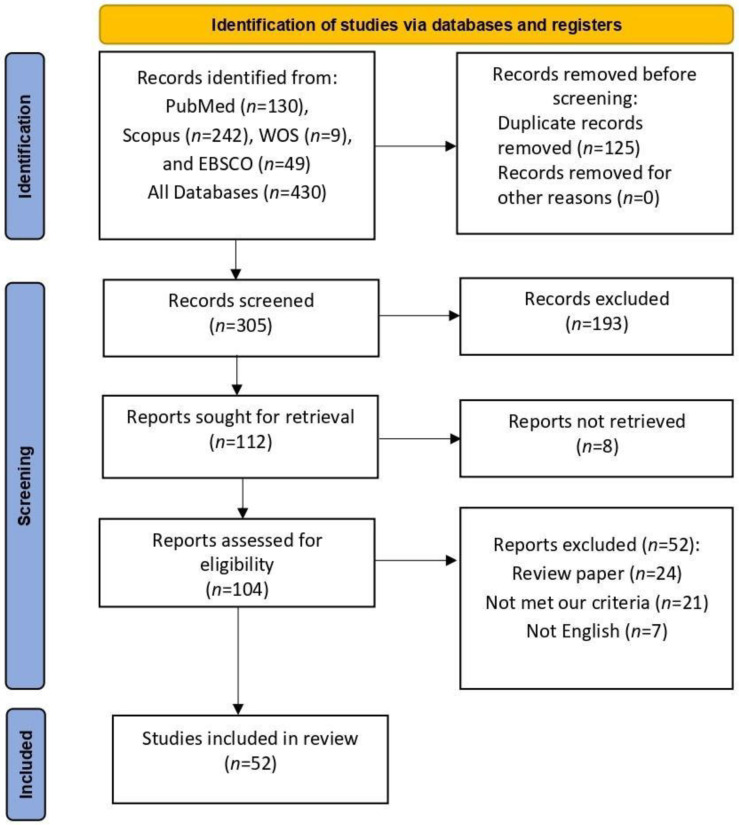
Shows the PRISMA flow diagram of the included studies.

**Figure 2 vaccines-11-00236-f002:**
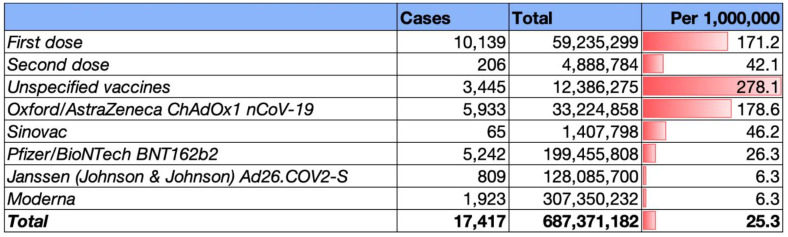
Represents the frequency of Bell’s palsy per 1,000,000 individuals after COVID-19 vaccination.

**Table 1 vaccines-11-00236-t001:** Characteristics of the patients.

		Frequency (*n*)	Percentage (%)
Characteristics (*n* = 105)			
Age, mean (SD)	49.7 (23.2)		
Sex	Male	60	57.1
	Female	45	42.9
Comorbidities			
Chronic kidney disease	NA	11	10.5
	No	93	88.6
	Yes	1	0.9
HIV	NA	11	10.5
	No	93	88.6
	Yes	1	0.9
Diabetes	NA	49	46.7
	No	39	37.1
	Yes	17	16.2
Hypertension	NA	46	43.8
	No	33	31.4
	Yes	26	24.8
Dyslipidemia	NA	11	10.5
	No	91	86.7
	Yes	3	2.8
Asthma	NA	11	10.5
	No	92	87.6
	Yes	2	1.9
Migraine	NA	11	10.5
	No	93	88.6
	Yes	1	0.9
Hyperuricemia	NA	11	10.5
	No	93	88.6
	Yes	1	0.9
Obesity	NA	11	10.5
	No	93	88.6
	Yes	1	0.9
Heart failure	NA	59	56.2
	No	41	39
	Yes	5	4.8
Thyroid dysfunction	NA	11	10.5
	No	93	88.6
	Yes	1	0.9
Cancer	NA	11	10.5
	No	93	88.6
	Yes	1	0.9
Poliomyelitis	NA	11	10.5
	No	93	88.6
	Yes	1	0.9
Stroke	NA	11	10.5
	No	93	88.6
	Yes	1	0.9
Obstructive sleep apnea	NA	11	10.5
	No	93	88.6
	Yes	1	0.9
Meniere’s disease	NA	11	10.5
	No	93	88.6
	Yes	1	0.9
Vaccines			
Type of given vaccine	NA	1	0.9
	Moderna	6	5.7
	Pfizer	49	46.7
	AstraZeneca	16	15.3
	Sinovac Biotech	22	21
	Johnson & Johnson	2	1.9
	Sputnik V	7	6.7
	Vector based vaccine	1	0.9
	COVAXIN	1	0.9
Doses			
Dosage resulted in the symptoms	NA	57	54.3
	1	37	35.2
	2	11	10.5
Diagnostic tools, treatments, and prognosis (*n* = 52)			
Diagnostic tools			
Physical examination	NA	1	1.9
	No	2	3.8
	Yes	49	94.2
CT	No	44	84.6
	Yes	8	15.4
MRI	No	23	44.2
	Yes	29	55.8
EMG	No	46	88.5
	Yes	6	11.5
Nerve conduction	No	47	90.4
	Yes	5	9.6
Lumbar puncture	No	31	59.6
	Yes	21	40.4
Electrophoresis	NA	1	1.9
	No	39	75
	Yes	12	23.1
Treatments			
Prednisone	NA	11	21.2
	No	21	40.4
	Yes	20	38.5
Methylprednisolone	NA	11	21.2
	No	38	73.1
	Yes	3	5.8
Deflazacort	NA	11	21.2
	No	40	76.9
	Yes	1	1.9
IVIG	NA	11	21.2
	No	27	51.9
	Yes	14	26.9
Anti-viral	NA	11	21.2
	No	34	65.4
	Yes	7	13.5
Plasma exchange	NA	11	21.2
	No	40	76.9
	Yes	1	1.9
Follow-up time, mean (SD)	30.19 (72.55)		
Outcome			
The affected side of the face	Unilateral	36	69.2
	Bilateral	16	30.8
Time of symptoms onset (days), mean (SD)	11.6 (11.13)		
Prognosis	NA	12	21.2
	No response	5	9.6
	Partial response	13	25
	Complete response	23	44.2
Overall prognosis	NA	12	21.2
	Not Recovered	5	9.6
	Recovered	36	69.2

**Table 2 vaccines-11-00236-t002:** Represents the possible factors affecting the prognosis of the COVID-19 vaccine-induced Bell’s palsy (*n* = 40).

Characteristics		Not Recovered *n* (%)	Recovered *n* (%)
Gender			
	Male	3 (11.5)	23 (88.5)
	Female	1 (7.1)	13 (92.9)
Age			
	<40	1 (6.7)	14 (93.3)
	≥40	3 (12)	22 (88)
Affected side			
	Unilateral	1 (4.2)	23 (95.8)
	Bilateral	3 (18.8)	13 (81.3)
Doses			
The dose that led to the symptoms			
	NA	1 (25)	3 (75)
	First	3 (10)	27 (90)
	Second	0 (0.0)	6 (100)
Vaccines			
Type of vaccine			
	NA	0 (0.0)	1 (100)
	Moderna	0 (0.0)	5 (100)
	Pfizer	1 (8.3)	11 (91.7)
	AstraZeneca	2 (18.2)	9 (81.8)
	Sinovac Biotech	0 (0.0)	1 (100)
	Johnson & Johnson	0 (0.0)	1 (100)
	Sputnik V	1 (14.3)	6 (85.7)
	Vector based vaccine	0 (0.0)	1 (100)
	COVAXIN	0 (0.0)	1 (100)
Comorbidities			
Chronic Kidney disease			
	NA	2 (18.2)	9 (81.8)
	No	2 (7.1)	26 (92.9)
	Yes	0 (0.0)	1 (100)
HIV			
	NA	2 (18.2)	9 (81.8)
	No	2 (7.1)	26 (92.9)
	Yes	0 (0.0)	1 (100)
Diabetes			
	NA	2 (18.2)	9 (81.8)
	No	2 (7.1)	26 (92.9)
	Yes	0 (0.0)	1 (100)
Hypertension			
	NA	2 (18.2)	9 (81.8)
	No	0 (0.0)	25 (100)
	Yes	2 (50)	2 (50)
Dyslipidemia			
	NA	2 (18.2)	9 (81.8)
	No	2 (7.1)	26 (92.9)
	Yes	0 (0.0)	1 (100)
Asthma			
	NA	2 (18.2)	9 (81.8)
	No	2 (7.1)	26 (92.9)
	Yes	0 (0.0)	1 (100)
Migraine			
	NA	2 (18.2)	9 (81.8)
	No	2 (7.1)	26 (92.9)
	Yes	0 (0.0)	1 (100)
Hyperuricemia			
	NA	2 (18.2)	9 (81.8)
	No	1 (3.6)	27 (96.4)
	Yes	1 (100)	0 (0.0)
Obesity			
	NA	2 (18.2)	9 (81.8)
	No	2 (7.1)	26 (92.9)
	Yes	0 (0.0)	1 (100)
Heart disease			
	NA	2 (18.2)	9 (81.8)
	No	2 (6.9)	27 (93.1)
Thyroid dysfunction			
	NA	2 (18.2)	9 (81.8)
	No	2 (7.1)	26 (92.9)
	Yes	0 (0.0)	1 (100)
Cancer			
	NA	2 (18.2)	9 (81.8)
	No	2 (6.9)	27 (93.1)
Poliomyelitis			
	NA	2 (18.2)	9 (81.8)
	No	2 (7.1)	26 (92.9)
	Yes	0 (0.0)	1 (100)
Stroke			
	NA	2 (18.2)	9 (81.8)
	No	2 (6.9)	27 (93.1)
Obstructive sleep apnea			
	NA	2 (18.2)	9 (81.8)
	No	2 (6.9)	27 (93.1)
Meniere’s disease			
	NA	2 (18.2)	9 (81.8)
	No	2 (6.9)	27 (93.1)
Treatments			
Prednisone			
	NA	0 (0.0)	2 (100)
	No	3 (15)	17 (85)
	Yes	1 (5.6)	17 (94.4)
Methylprednisolone			
	NA	0 (0.0)	2 (100)
	No	4 (11.4)	31 (88.6)
	Yes	0 (0.0)	3 (100)
Deflazacort			
	NA	0 (0.0)	2 (100)
	No	4 (10.8)	33 (89.2)
	Yes	0 (0.0)	1 (100)
IVIG			
	NA	0 (0.0)	2 (100)
	No	2 (8.3)	22 (91.7)
	Yes	2 (14.3)	12 (85.7)
Anti-viral			
	NA	0 (0.0)	2 (100)
	No	4 (12.5)	28 (87.5)
	Yes	0 (0.0)	6 (100)
Plasma exchange			
	NA	0 (0.0)	2 (100)
	No	4 (10.8)	33 (89.2)
	Yes	0 (0.0)	1 (100)

**Table 3 vaccines-11-00236-t003:** The quality assessment of the included case reports using the (JBI) Quality Assessment Tool.

Study ID	The Joanna Briggs Institute (JBI) Critical Appraisal Checklist for Case Reports								Total Quality Score, (%)
	Q (1)	Q (2)	Q (3)	Q (4)	Q (5)	Q (6)	Q (7)	Q (8)	
Burrows 2021 [16]	YES	YES	YES	YES	YES	YES	NO	YES	88%
Cellina 2022 [17]	YES	YES	YES	YES	YES	YES	NO	YES	88%
Čenščák 2021 [18]	YES	YES	YES	YES	YES	YES	NO	YES	88%
Cirillo 2022 [19]	NO	NO	YES	YES	YES	YES	NO	YES	62.50%
Colella 2021 [20]	YES	YES	YES	NO	YES	YES	NO	YES	75%
Ercoli 2021 [2]	YES	YES	YES	YES	YES	YES	NO	YES	88%
Finsterer 2021 [21]	YES	YES	YES	YES	YES	YES	NO	YES	88%
Iftikhar 2021 [22]	YES	YES	YES	YES	YES	YES	NO	YES	88%
Introna 2021 [23]	YES	YES	YES	YES	YES	YES	NO	YES	88%
Ish 2021 [24]	YES	NO	YES	NO	YES	YES	NO	YES	62.50%
Manea 2021 [25]	YES	YES	YES	YES	YES	YES	NO	YES	88%
Martin-Villares 2021 [26]	NO	YES	YES	YES	NO	YES	NO	YES	75%
Mason 2021 [27]	YES	NO	YES	YES	YES	YES	NO	YES	75%
Mussatto2021 [28]	YES	NO	YES	YES	YES	YES	NO	YES	75%
Nasuelli 2021 [29]	YES	NO	YES	YES	YES	YES	NO	YES	75%
Nishizawa 2021 [30]	YES	YES	YES	YES	NO	NO	NO	YES	62.50%
Obermann 2021 [31]	YES	YES	YES	YES	YES	YES	NO	YES	88%
Pothiawala 2021 [32]	YES	YES	YES	NO	YES	NO	NO	YES	62.50%
Poudel 2022 [33]	YES	NO	YES	YES	YES	YES	NO	YES	75%
Repajic 2021 [34]	YES	YES	YES	NO	YES	YES	NO	YES	75%
Shalabi 2022 [35]	YES	NO	YES	YES	YES	YES	NO	YES	75%
Yu 2021 [36]	YES	NO	YES	YES	YES	YES	NO	YES	75%
Zhang 2022 [37]	YES	NO	YES	NO	YES	YES	NO	YES	62.50%

## Data Availability

Not applicable.

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
