# Peer review of "Bell’s Palsy and COVID-19 Vaccines: A Systematic Review and Meta-Analysis"

_vaccines, 2023, doi:10.3390/vaccines11020236_

Round 1

Reviewer 1 Report

Bell's palsy is a disorder due to unilateral weakness, malfunction of facial muscles. It is typically transitory condition which improves after a few weeks and full recovery is described in about six months – and commonly caused by inflammation of the nerve that innervates the muscles on one side of the face. One of the most common etiological factors of the disorder is an earlier viral infection. Authors pointed out those vaccines against SARS-CoV-2 generally play a critical role to decrease the COVID-19 rate and severity, as well as save a lot of people worldwide from consequences or lethal outcome of this disease. However, they were mentioned some side effects for these vaccines like pain, fatigue, fever, and neurological defects (including Bell’s palsy). In this systematic review (using data from PubMed, SCOPUS, EBSCO, and Web of Science), the authors aimed to collect evidence for the occurrence and incidence of Bell's palsy after vaccination against SARS-CoV-2 using different (Pfizer, Biotech, Moderna, AstraZeneca, Johnson & Johnson, Sputnik V, etc) vaccines. Authors accept (confirm by citing) and consider a rising rate of different neurological (and other) complications/disorders following anti SARS-CoV-2 vaccine administrations. Is it an essential and realistic fact or viewpoint, considering the beneficial effects of vaccination on the common population?! In addition, as mentioned "The exact mechanism of developing facial nerve palsy following COVID-19 vaccination is yet to be unclear…. One theory suggests the involvement of autoimmune processes (via mimicry of host molecules by the vaccine antigens or by bystander activation of autoreactive dormant T-cells)…. Another theory states that the reactivation of different dormant viruses in the CNS might cause the palsy  ……".

In their conclusion, authors stated that the incidence of Bell's palsy after anti SARS-CoV-2 vaccination was practically insignificant, only 25/1000000 (0.0025 percent). Certainly, the number of subjects examined is too small, and the benefit from vaccination is undeniably greater than the potential risk. I am of the opinion that there would be no particular benefit from the publication of a paper examining the association between Bell's palsy and vaccinations against SARS-CoV-2.

Author Response

General comments: we thank the peer-reviewers for their prompt and positive criticism of our manuscript. We believe our manuscript is now more robust and in excellent shape to proceed with publication. Our point-by-point responses are found below:

Reviewer (1):

Reviewer question: I am of the opinion that there would be no particular benefit from the publication of a paper examining the association between Bell's palsy and vaccinations against SARS-CoV-2.

The authors reply: Thank you for your comment. Up-to-date, there is no data that pooled all medical citations to investigate the occurrence and characteristics of bell’s palsy after COVID-19 vaccination. Furthermore, this investigation were included 52 citations and represents, to the best of our knowledge, the most updated and largest number of included studies. Moreover, we tried in the DISCUSSION section to reveal the gap between studies and provide more information about the strengths and limitations of our review

Reviewer 2 Report

I read with great interest the paper of Albakri and colleagues. However, I strongly suggest for it to be revised following the comments below:

1. Please revise figure 1 because it seems that it was distorted in the formatted manuscript. Better incorporate it in the text as JPEG or PNG file.

2. The methods section is haphazardly done. Please include the online registry number for this study. Kindly define as well the P.I.C.O and follow the PRISMA format. Please include too a detailed description of the inclusion and exclusion criteria for the articles selected.

3. For the data analysis, where is the forest plot? Please include it.

4. For the quality assessment using Joanna Briggs, better if you generate a heat map for presentation.

5. It should be emphasized too in the introduction that several studies were published already pertaining to the effects of COVID-19 vaccines. Kindly cite the following articles:

doi: 10.47176/mjiri.36.85
doi: 10.1016/j.eclinm.2022.101624 doi: 10.1016/S1473-3099(21)00467-9

Author Response

General comments: we thank the peer-reviewers for their prompt and positive criticism of our manuscript. We believe our manuscript is more robust and in excellent shape to proceed with publication. Our point-by-point responses are found below:

Reviewer (2):

Reviewer question: 1. Please revise figure 1 because it seems that it was distorted in the formatted manuscript. Better incorporate it in the text as JPEG or PNG file.

The authors reply: Thank you for your comment and sorry for this technical mistake. We adjusted “FIGURE 1” according to your suggestion.

Reviewer question: 2. The methods section is haphazardly done. Please include the online registry number for this study. Kindly define as well the P.I.C.O and follow the PRISMA format. Please include too a detailed description of the inclusion and exclusion criteria for the articles selected.

The authors reply: Thank you for this valuable comment. The METHODS section was adjusted according to your suggestions.

Reviewer question: 3. For the data analysis, where is the forest plot? Please include it.

The authors reply: Thank you for your comment. Unfortunately, there is no FOREST PLOT was generated, because we pooled groups' and individuals’ data and we performed the analysis in SPSS software.

Reviewer question: 4. For the quality assessment using Joanna Briggs, better if you generate a heat map for presentation.

The authors reply: Thank you for your comment. See TABLE 3. For Joanna Briggs heat map and summary.

Reviewer question:  5. It should be emphasized too in the introduction that several studies were published already pertaining to the effects of COVID-19 vaccines. Kindly cite the following articles: (doi: 10.47176/mjiri.36.85), (doi: 10.1016/j.eclinm.2022.101624), (doi: 10.1016/S1473-3099(21)00467-9).

The authors reply: Thank you for your comment. We added the following sentences in the INTRODUCTION section comment on conclusions of published study regarding this topic Recent studies suggested further evaluation of bell’s palsy and COVID-19 vaccination, and till now there was no strong evidence can confidently conclude whether there is an association between bell’s palsy and COVID-19 vaccinations or not. Furthermore, a systematic review of 17 studies was published to investigate the existence and development of COVID-19 symptoms after vaccinations. They concluded that COVID-19 vaccination before SARS-COV-2 infection could reduce the risk of long-term symptoms of COVID-19.”

Round 2

Reviewer 1 Report

Vaccination is undoubtedly essential for prevention of infection with SARS-CoV-2, to improve COVID-19 clinical course/outcome and to decrease of disease related mortality. There are some generally accepted potential side or adverse effects for different anti SARS-CoV-2 vaccines. As authors mentioned, pain, fatigue, fever, and different immuno-neurological defects, such as Guillain-Barre syndrome, polyneuritis or some other forms of autoimmune neuropathy, including Bell’s palsy too are the most frequent side effects or complications.

Authors presented that the percentage of the Bell’s palsy after COVID-19 vaccination (using different vaccines) was very low, but existing (approximately 0.00253%). They also observed that the ratio of this side effect (Bell’s palsy) was higher after first than following second dose of vaccines and it was the highest for Oxford/AstraZeneca vaccine compared to other vaccines included in this data analysis. 

After the corrections, the revised version of this manuscript could be accepted for publication, despite the low frequency of this complication and the examined database. Ps I did not notice any cover letter in the submitted material. However, I agree that the paper should be published in this form.

Reviewer 2 Report

The paper has been revised extensively and can now be published.